# Influence of Resveratrol on the Cardiovascular Health Effects of Chronic Kidney Disease

**DOI:** 10.3390/ijms21176294

**Published:** 2020-08-31

**Authors:** Jenn-Yeu Song, Ta-Chung Shen, Yi-Chou Hou, Jia-Feng Chang, Chien-Lin Lu, Wen-Chih Liu, Po-Jui Chen, Bo-Hau Chen, Cai-Mei Zheng, Kuo-Cheng Lu

**Affiliations:** 1Division of Cardiovascular Surgery, Department of Surgery, Taipei Tzu Chi Hospital, Buddhist Tzu Chi Medical Foundation, New Taipei City 231, Taiwan; doc98017@gmail.com (J.-Y.S.); frederick_shen@hotmail.com (T.-C.S.); 2School of Medicine, Tzu Chi University, Hualien 970, Taiwan; 3Division of Nephrology, Department of Medicine, Cardinal-Tien Hospital, School of Medicine, Fu Jen Catholic University, New Taipei City 234, Taiwan; athletics910@gmail.com; 4Division of Nephrology, Department of Internal Medicine, En Chu Kong Hospital, New Taipei City 237, Taiwan; cjf6699@gmail.com; 5Division of Nephrology, Department of Medicine, Fu Jen Catholic University Hospital, School of Medicine, Fu Jen Catholic University, New Taipei City 242, Taiwan; janlin0123@gmail.com; 6Division of Nephrology, Department of Medicine, Taipei Hospital, Ministry of Health and Welfare, New Taipei City 242, Taiwan; wayneliu55@gmail.com; 7Department of Pediatrics, Taoyuan Armed Forces General Hospital, Taoyuan City 325, Taiwan; mythnobody0221@gmail.com (P.-J.C.); dreamvenice@hotmail.com (B.-H.C.); 8Taipei Medical University-Research Center of Urology and Kidney, Taipei Medical University, Taipei 110, Taiwan; 9Division of Nephrology, Department of Internal Medicine, Taipei Medical University Shuang Ho Hospital, New Taipei City 235, Taiwan; 10Division of Nephrology, Department of Internal Medicine, School of Medicine, College of Medicine, Taipei Medical University, Taipei 110, Taiwan; 11Division of Nephrology, Department of Medicine, Taipei Tzu Chi Hospital, Buddhist Tzu Chi Medical Foundation, New Taipei City 231, Taiwan; kuochenglu@gmail.com

**Keywords:** cardiovascular disease, chronic kidney disease, indoxyl sulfate, microbiota, *p*-cresol sulfate, resveratrol, trimethylamine-N-oxide

## Abstract

Cardiovascular disease (CVD) is closely related to chronic kidney disease (CKD), and patients with CKD have a high risk of CVD-related mortality. Traditional CVD risk factors cannot account for the higher cardiovascular risk of patients with CKD, and standard CVD interventions cannot reduce the mortality rates among patients with CKD. Nontraditional factors related to mineral and vitamin-D metabolic disorders provide some explanation for the increased CVD risk. Non-dialyzable toxins, indoxyl sulfate (IS) and *p*-cresol sulfate (PCS)—produced in the liver by colonic microorganisms—cause kidney and vascular dysfunction. Plasma trimethylamine-N-oxide (TMAO)—a gut microbe-dependent metabolite of dietary L-carnitine and choline—is elevated in CKD and related to vascular disease, resulting in poorer long-term survival. Therefore, the modulation of colonic flora can improve prospects for patients with CKD. Managing metabolic syndrome, anemia, and abnormal mineral metabolism is recommended for the prevention of CVD in patients with CKD. Considering nontraditional risk factors, the use of resveratrol (RSV), a nutraceutical, can be helpful for patients with CVD and CKD. This paper discusses the beneficial effects of RSV on biologic, pathophysiological and clinical responses, including improvements in intestinal epithelial integrity, modulation of the intestinal microbiota and reduction in hepatic synthesis of IS, PCS and TMAO in patients with CVD and CKD.

## 1. Introduction

Cardiovascular disease (CVD) is the leading cause of death in patients with chronic kidney disease (CKD). The strong causality between CKD and CVD risk means that preventing the progression of CKD can also prevent CVD. In patients with CKD, increased CVD risk is multifactorial, and a targeted intervention for a single traditional risk factor is inadequate. Therefore, research on innovative therapeutic strategies and CVD risk factors is essential [1]. We evaluated traditional and nontraditional risk factors of CVD in CKD and analyzed their relationship. Nutraceuticals are derived from foods and have been demonstrated to have physiological benefits, establishing them as likely the most suitable option for preventing CVD in patients with CKD.

Resveratrol (RSV), a nutraceutical, is a naturally occurring polyphenol that can be found in red wine and grapes [2]. Reports have indicated that it is beneficial for treating or preventing CVD by improving metabolic syndrome components [3,4]. Moreover, RSV reduces the amount of plasma protein-bound uremic toxins and trimethylamine-N-oxide (TMAO) levels by maintaining balance in the gut microbiota [5]. RSV has been demonstrated to be safe and tolerable in humans. This overview focuses on how RSV can influence traditional and nontraditional cardiovascular risk factors and provide beneficial effects by attenuating the effect of uremic toxins in patients with CKD.

## 2. Relationship between CVD and CKD

The traditional risk factors common for CVD and CKD are advanced age, hypertension, diabetes, dyslipidemia, smoking habit, family history of CVD or CKD and male sex. Toxic metabolites produced during uremia, such as indoxyl sulfate (IS), account for most cases of CVD in patients with CKD. Changes in how these chemicals are metabolized constitute nontraditional risk factors [6,7]. Other nontraditional factors, including TMAO, accumulate in patients with CKD and aggravate CVD. In addition, dysregulation of calcium, phosphorus, PTH or vitamin D metabolism was evaluated.

Screening and testing in the early stages of CKD can aid the development of interventions that can delay disease progression. In fact, the focus of treatments and interventions should be shifted to the early stage of CKD because early identification through screening may substantially attenuate the impact of CKD and delay or even prevent its development [8].

### 2.1. Oxidative Stress in Chronic Kidney Disease

Oxidative stress (OS) is caused by the overproduction of reactive oxygen/nitrogen species (RONS) and weakened antioxidant defense and may cause blood vessel and tissue damage and DNA damage in patients with CKD [9]. Patients with CKD who have increased RONS production, impaired nonenzymatic or enzymatic antioxidant defense, increased uremic toxins (IS), severe isoform cystine deficiency, insufficient biocompatibility of dialysis-related membranes and the inhibition of normal cell function by endotoxins [9]. The uncoupling of uremic toxin-induced endothelial nitric oxide synthase (eNOS) [10], increased activity of nicotinamide adenine dinucleotide phosphate-oxidases [11,12], and loss of antioxidants due to insufficient dietary intake and reduced intestinal absorption [13] contribute to OS in CKD.

Chronic inflammation and OS are intrinsically linked because they mutually enhance each other. Nuclear factor κB (NFκB), a redox-sensitive transcription factor, regulates proinflammatory cytokine and chemokine production. OS can promote the activation of leukocytes and resident cells, thereby initiating inflammatory response [14]. Renal mitochondria are rich in oxidative reactions and are therefore susceptible to OS. Evidence suggests that OS can accelerate the progression of renal function. In advanced CKD, elevated OS is associated with numerous metabolic complications [15]. This results in the formation of a vicious cycle, which ultimately leads to a high death rate in patients with CKD [16].

### 2.2. Inflammation in Chronic Kidney Disease

Although CKD is related to systemic inflammation, many other factors may cause CKD. Recently, persistent low-grade inflammation has been identified as a critical component of CKD that plays a unique role in its pathophysiology and, to a certain extent, leads to the development of cardiovascular disease, all-cause death and wastage of protein energy [17]. In mice with progressive CKD associated with Alport syndrome, the production of intestinal bacteriocins is unregulated, live bacteria are transported through the gut barrier into the liver, and levels of microbial endotoxins increase in the serum [18]. Therefore, uremia is related to enteral malnutrition, gut barrier dysfunction and microbial translocation, all of which cause persistent systemic inflammation in CKD [19].

Cohen et al. described patients with heart disease who had a proinflammatory patterns of high levels of IL-1, IL-6 and TNF-α combined with low anti-inflammatory parameters—including IL-2, IL-4, IL-5 and T cell number. The patients with high proinflammatory cytokines had a lower survival rate than patients without a characteristic cytokine pattern [20]. In addition, external factors such as impurities in dialysis water, quality of the dialysate and biologically incompatible factors in the dialysis circuit, act as dialysis-related factors [21]. On a histological level, regardless of initial injury, inflammation resulted in renal fibrosis. Mounting evidence suggests that renal inflammation plays a central role in the development and progression of CKD. Therefore, restoring balance between profibrotic and antifibrotic signaling pathways can be used as a strategy to develop antifibrotic strategies that target several pathways simultaneously [22].

## 3. Effect of Nontraditional Risk Factors (Uremic Toxins) on the Development of Cardiovascular Disease

In patients with CKD, CVD is highly prevalent and traditional risk factors cannot adequately predict cardiovascular events. In CKD, the influx of urea and other residual toxins alters the gut microbiota. The number of beneficial microbes that produce short-chain fatty acids (epithelial energy source) decreases, whereas the number of microbes that produce uremic toxins increases [23].

Indoxyl sulfate and *p*-cresol sulfate (PCS) accumulate in the organs of patients with CKD. The local and systemic disturbances caused by the toxins in biologic metabolism and cellular signal transduction result in uremic syndrome. Chronic kidney disease perturbs inter-organ communication through small molecules, including uremic solutes and signaling molecules [24]. Moreover, IS plays a role in regulating drug-metabolizing enzymes (DMEs) and transporters during inter-organ communication [25]. In healthy individuals, the concentration of IS is 0.1–2.39 μM in healthy individuals but exceeds 500 μM in patients with advanced CKD [26]. In addition, IS is related to the progression of CVD in patients with CKD, which may be due to increased OS in the myocardium and vasculature [27]. Barreto et al. demonstrated that IS may be related to a higher CVD mortality in patients with CKD [28].

TMAO is derived from the metabolites of the gut microbiota, dietary choline, lecithin and L-carnitine. In patients with CKD, serum TMAO levels are high, which contributes to lower survival. In a mouse model, TMAO accelerated atherosclerosis by inhibiting cholesterol transfer from tissues to the liver [29]. Studies have shown that TMAO is independently related to cardiovascular events and that a relationship between TMAO levels and ischemic cardiovascular events exists, especially in CKD Stages 3b and 4 [30]. For these reasons, selective inhibition of TMAO prevents renal damage and cardiovascular events in patients with CKD [31].

### 3.1. RSV Improves Intestinal Epithelial Integrity

RSV Enhances Intestinal Epithelial Integrity, Shapes Microbiota and Reduces the Synthesis of Indoxyl Sulfate in the Liver (Figure 1)

The main functions of intestinal mucosa are nutrient absorption, waste secretion and the prevention of waste absorption [32]. The apical junctional complex, which includes the tight junction (TJ) and the adherens junction, prevents intercellular permeation of intestinal contents [33]. The TJ consists of different transmembrane and cytosolic components [33,34]. Transmembrane proteins include occludin, claudin and junctional adhesion molecule-A; cytosolic proteins include members of the zonula occludens (ZO) protein family, of which ZO-1 plays a crucial role in TJ assembly and function [33]. CKD causes the disintegration and reduction of colonic TJ proteins [35].

Urolithin A (UroA), derived from polyphenolics, demonstrates anti-inflammatory and antioxidative activities [36]. Urolithin A and its effective synthetic analog (UAS03) improve intestinal barrier function and inflammation significantly. UroA and UAS03 activate aryl hydrocarbon receptor/Nrf2-dependent pathways to increase the production of epithelial TJ proteins [37]. RSV is one of the most researched natural polyphenols and may enhance intestinal epithelial integrity through the increased production of epithelial TJ proteins [38]. It has also been shown that RSV increases the expression of TJ, desmosomes in colorectal cancer cells. Thus, the epithelization is an important property of RSV, even in tumor cells [39].

#### 3.1.1. RSV Shapes Intestinal Microbiota

RSV modulates the gut microbiota to reduce body weight and fat and improve glucose metabolism and obesity-related indices [40]. Moreover, it increases the abundance of beneficial probiotics, but reduces the growth of *Enterococcus faecalis* [41,42], which generates oxygen radicals that damage colonic epithelial cell DNA. RSV supplementation can increase butyrate-producing microbes that alleviate OS [43]. Furthermore, polyphenols inhibit the growth of TMAO-producing bacteria, thereby reducing TMAO levels [44].

In the colon, the bacterial metabolites of polyphenols can affect microbial composition and function [45]. Studies suggest that either shaping the intestinal microbiota through specific bacterial species or reducing the Firmicutes/Bacteroidetes ratio can provide crucial benefits for the host [46].

Moreover, RSV attenuates TMAO-induced atherosclerosis by reducing TMAO levels through shaping of the gut microbiota [42]. Moreover, RSV increases bile salt hydrolase (BSH) activity, which promotes the generation of unconjugated bile acid (BA) and enhances fecal BA loss. Fecal BA loss leads to increased hepatic CYP7A1 expression, thereby inducing hepatic BA synthesis, which reduces hepatocyte and plasma cholesterol levels and subsequently attenuates atherosclerosis [47].

The deletion of sirtuin-1 (SIRT-1) in the gut epithelium reduces the levels of anti-inflammatory *Lactobacillus*. RSV activates intestinal epithelial SIRT1 by regulating the gut microbiota, thereby preventing intestinal inflammation [48].

#### 3.1.2. RSV Reduces Hepatic Synthesis of Indoxyl Sulfate, *p*-Cresol Sulfate and Trimethylamine-*N*-oxide

Tryptophan, derived from dietary protein, is first metabolized into indole by the tryptophanase of enteric bacteria [49]. Thereafter, indole is absorbed in the intestine, converted into indoxyl by cytochrome P450 (CYP2E1, CYP2C19, CYP2A6) and coupled to IS by sulfotransferase (SULT) in the liver [50]. Indoxyl sulfate, which is mainly produced in the liver, then enters the circulatory system. Before being excreted through urine, IS is absorbed through organic anion transporters, OAT1 and OAT3, localized in the renal proximal tubular cells [51,52]. Moreover, IS production is blocked through the suppression of SULT through RSV in the rat liver S9 fraction [52].

Gut microbiota dysbiosis results in the development of proteolytic fermentation through the growth of bacteria that produce uremic toxins (IS, PCS and TMAO). Oral RSV can alter the composition of the intestinal microbiota to reduce TMA production, then the hepatic synthesis of TMAO and finally plasma TMAO levels [47]. Thus, RSV can reduce the serum levels of IS or PCS through the inhibition of hepatic SULT and reduce TMAO levels by reducing the gut microbial production of TMA.

### 3.2. Signaling Pathway Involved in Inflammation and OS

Pharmacokinetic analysis revealed that RSV is rapidly metabolized in the body. Although 70% of RSV is absorbed, its bioavailability after oral intake is low [53]. The beneficial effects of RSV may be due to its ability to protect the body from reactive oxygen species (ROS) injury [54], suppression of cyclooxygenase [55] or activation of anti-inflammatory signaling pathways such as the SIRT-1 pathway [56]. SIRT-1 inhibits TLR4/nuclear factor κB (NF-κB)/STAT signaling, which reduces the production of cytokines [57] and macrophage-derived proinflammatory factors [58].

#### 3.2.1. Epithelial Nitric Oxide Synthase/Inducible Nitric Oxide Synthase Balance

NO synthesis occurs through the nitric oxide synthase (NOS)-dependent or NOS-independent pathways. The reduction of nitrite into NO mainly occurs in the NOS-independent pathway [59]. NOS catalyzes the oxidation of L-arginine to L-citrulline, which is then used to synthesize NO.

Neuronal NOS and eNOS are mainly expressed during physiological processes; however, inducible NOS (iNOS) is more likely to be expressed in a pathologic state [60]. Depletion of L-arginine or cofactor tetrahydrobiopterin and the uncoupling of eNOS increase superoxide production [60]. The increase in OS could lead to vascular endothelial dysfunction.

The vascular endothelium determines aortic permeability for macromolecules and leukocytes, regulates vascular tone and prevents coagulation. Endothelial cell–derived NO reduces smooth muscle cell (SMC) contractility, which regulates vascular tone [61]. In a Marfan syndrome mouse model, NO synthesis was impaired [61]. Endothelium dysfunction is defined as either the loss or overproduction of NO [62]. Excess production of NO, driven by iNOS, aggravates OS and cellular damage through the accumulation of peroxynitrites [63]. For this reason, RSV has been identified as the trigger for NO synthesis in endothelial cells [64].

Studies have demonstrated that RSV increases the expression of Krüppel-like factor 2 (KLF2) [65] through the activation of SIRT-1 [66] and production of eNOS [67,68,69], thereby promoting endothelial cell function. Studies on rats with STZ-induced diabetes have revealed that RSV accentuates endothelial dysfunction through the attenuation of OS, thereby enhancing the expression of NOS3, enriching NO bioavailability and diminishing transforming growth factor β expression [70]. Notably, overexpression of endothelial microRNA-21 (miR-21) induces eNOS production; thus, the induction of miR-21 by RSV likely plays a crucial role in preventing vascular endothelial dysfunction [71].

#### 3.2.2. NADPH Oxidase 4 and Reactive Oxygen Species

The biologic function of NADPH oxidase (NOX) involves the generation of reactive oxygen species [72] through electron transfer across biologic membranes. NOX-dependent ROS production causes persistent OS, eNOS uncoupling and poor mitochondrial function, thereby increasing ROS production and causing tissue injury, which is related to the progression of CVD [11].

The antioxidant characteristic of RSV is likely attributable to its role as a gene regulator. RSV inhibits NOX-mediated production of ROS by downregulating gene expression and thus the activity of the oxidase. Moreover, RSV attenuates mitochondrial superoxide production and increases the expression of various antioxidant enzymes; some gene-regulating effects of RSV are mediated by histone/protein deacetylase SIRT-1 or by nuclear factor E2-related factor 2 [6].

#### 3.2.3. NAD^+^-Dependent Protein Deacetylase Sirtuin-1 Activation

SIRT-1 deacetylases various transcription factors that are involved in lifespan prolongation. The expression of NO synthase is upregulated by SIRT-1, which also upregulates fork head box O to maintain vascular endothelial morphology [73]. RSV also augments eNOS expression through SIRT-1 activation. Moreover, the upregulation of SIRT-1 and eNOS in response to calorie restriction has been reported [74]. Over expression of SIRT-1 leads to an increase in eNOS expression [75]. In coronary arterial endothelial cells, SIRT-1 knockdown attenuates RSV-induced eNOS expression [76]. Fourny et al. showed that RSV ameliorates ischemia-reperfusion injury in the heart by increasing energy utilization and prolonging eNOS/SIRT-1 expression in female rats with type 2 diabetic [77].

#### 3.2.4. Heme Oxygenase

Heme oxygenase (HO), an antioxidant enzyme, modulates intracellular pro-oxidant heme, carbon monoxide and biliverdin, which have been reported to be upregulated during stress. Thus, to protect cells from stress, HO-1 promotes detrimental effects such as neurological disease and malignancy [78].

We have shown that HO-1 therapy attenuates the severity of membranous nephropathy (MN) through its antioxidative and immune modulatory effects [79]. Furthermore, we proved that RSV increased HO-1 expression and attenuated MN in a murine model. Therefore, RSV can potentially be used as a treatment for MN [80].

## 4. Effect of RSV on Traditional Cardiovascular Risk Factors

### 4.1. Effect of RSV on Vascular Function

Angiotensin II (Ang II) acts on the angiotensin II type 1 receptor (AT1R) to induce the production of ROS and subsequent OS. By contrast, angiotensin 1–7 (Ang 1–7) acts on the Mas receptor (MasR) and provides protective effects. Previous studies have shown that RSV can prevent renal function deterioration, ameliorate proteinuria and improve renal histological findings in aged mice by downregulating Ang II/AT1R activity and promoting Ang 1–7/MasR activity [81]. Another study demonstrated that RSV inhibits Ang II and increases the effects of angiotensin-converting enzyme 2 (ACE2) to prevent arterial aging [82]. RSV promotes the synthesis of nitric oxide (NO) from endothelial cells and suppresses the production of endothelin-1, thereby reducing OS. Mounting evidence suggests that RSV treatment protects against the detrimental effects of induced vascular damage. Thus, RSV has protective effects on vascular function and blood pressure [83].

### 4.2. Effect of RSV on Myocardial Function

Myocardial inflammation causes cardiac damage. Increased activation of NF-κB, a nuclear transcription factor responsible for the production of proinflammatory cytokines, is involved in the immune and inflammatory response of the myocardium, thereby aggravating myocardial injury [84].

In a rat model of sepsis, RSV was reported to inhibit related inflammatory factors and the NF-κB signaling pathway and activate the PI3K/mTOR signaling pathway, thereby protecting the myocardium during sepsis [85]. RSV therapy increases the expression of the autophagy biomarkers beclin-1 and LC-3II but reduces the expression of IL-6. These findings show that RSV has protective effects in ischemia-reperfusion injury in a diabetic rat model [86].

### 4.3. Effect of RSV on Metabolic Syndrome Components

Metabolic syndrome increases the risk of CVD, stroke and type II diabetes. RSV acts through different mechanisms to improve the symptoms of metabolic syndrome and related disorders [87]. RSV activates SIRT-1, which activates eNOS, thereby resulting in cardioprotective, antioxidant and anti-inflammatory effects [88,89]. Tamaki et al. reported that in addition to SIRT-1, RSV activated the adenosine monophosphate-activated kinase (AMPK) signaling and Nrf2/ARE antioxidant pathways in a rat periodontitis model [90]. In addition, RSV treatment markedly reduces profibrotic protein expression and increases the expression of the ACE2/MasR axis components to increase vasodilatation and reduce blood pressure. Moreover, RSV inhibits the migration of vessel SMCs, which have important antiatherogenic and antiatherosclerotic effects [82,91]. Thus, RSV plays an important role in metabolic syndrome (Figure 2).

#### 4.3.1. RSV and High Blood Pressure

The effects of RSV on blood pressure have been explored in numerous animal models [92,93,94]. RSV can reduce levels of serum Ang II and ACE. RSV treatment also increases serum Ang-(1e7) levels, which is accompanied by the increased expression of ACE2, angiotensin II type 2 receptor and MasR [82]. The blood pressure-lowering effects of RSV are attributable to the increased production of endothelial NO, which reduces vascular OS [83].

#### 4.3.2. Fat Accumulation and Cholesterolemia

RSV affects the lipid profile by increasing the expression of the cholesterol transporter protein. RSV enhances Apo-A1 synthesis and ameliorates foam cell formation through the PPAR-γ and adenosine 2A receptor pathways [95]. RSV enhances gut microbiota remodeling and BSH activity. In addition, RSV-induced fecal BA loss leads to increased expression of CYP7A1 in the liver, thereby inducing hepatic BA synthesis. Hepatic BA reduces the levels of hepatocyte and plasma cholesterol [47].

#### 4.3.3. RSV and Glucose Intolerance and Insulin Resistance

The effect of RSV on cellular glucose metabolism has been well studied. RSV improves insulin resistance (IR) in adipose tissue and the liver [96], improves hepatic IR by regulating long noncoding RNAs (lncRNAs) [97] and upregulating the miRNA mmu-miR-363-3p [98] and reduces hepatic endoplasmic reticulum stress, thereby improving insulin sensitivity [99].

In skeletal muscles, RSV promotes the phosphorylation of the α-subunit of AMPK to improve glucose metabolism [100] and increases glucose uptake by increasing the expression of the cell membrane localized glucose transporter type 4 [101,102,103]. RSV improves skeletal muscle fatty acid oxidation and reduces OS, thereby reducing IR [104]. Moreover, RSV ameliorates ROS levels in the muscle and liver cells in high-fat diet-induced IR [105].

## 5. Biologic Role of RSV in Atrial Fibrillation

Atrial fibrillation (AF) and CKD usually occur together, which poses a medical dilemma because of the risk of thromboembolism and bleeding episodes [106]. RSV directly affects heart function and rhythm through cardiac remodeling and ion channel activity [107]. RSV also suppresses hypertrophic heart remodeling through the activation of SIRT-1/AMPK and the subsequent inhibition of NFAT activation, which is implicated in the evolution of AF [108], cardiac myopathy and congestive heart failure [109]. In a study investigating the therapeutic efficacy of RSV in ameliorating AF in an animal model, RSV was found to attenuate atrial fibrosis and modulate ion channels to reduce AF through the PI3K/eNOS signaling pathway [110]. Thus, the cardiovascular protective effects of RSV include reducing OS and alleviating inflammation through Nrf2 and SIRT-1 activation, upregulating the PI3K/eNOS pathway and downregulating the NF-κB pathway [111].

## 6. RSV with Strong Anti-Carcinogenic Effect via Cardiovascular Protective Effects

Reactive oxygen species (ROS) play a pivotal role in the pathogenesis of both heart disease and tumor progression. Any disturbances in ROS metabolism results an increase in OS which initiates subcellular changes and resultant cardiomyopathy and heart failure. A previous study indicated that exogenous antioxidant RSV is of value in preventing both the development of heart disease and cancer by acting as ROS scavenger [112]. RSV reverses multidrug resistance in cancer cells and sensitizes cancer cells to standard chemotherapeutic agents. The proposed mechanisms of RSV to prevent carcinogenesis include the inhibition of OS, inflammation and cancer-cell proliferation and the activation of tightly regulated cell-death mechanisms [113]. RSV possesses a wide range of preventive and therapeutic options against different types of cancer [114] through its proapoptotic, antiproliferative and anti-inflammatory actions [115,116]. RSV also suppresses the malignant biologic behaviors of cancer cells, including proliferation, antiapoptosis, invasion, migration, EMT progress, levels of ROS and stemness [117]. Recently, RSV is proved be chemopreventive from tumorigenesis by targeting Sirt1 and suppression of NF-κB activation [118]. In brief, RSV provides additional anti-carcinogenic effects parallel with cardioprotective effects.

## 7. New Therapeutic Interventions

The poor bioavailability of RSV limits its use as a therapeutic drug. RSV has rapid Phase II metabolism in the liver and intestine [119,120] and sufficient blood levels cannot be reached through intravenous nor oral administration of RSV [121]. To overcome this challenge, the use of natural or synthetic analogs with better bioavailability or higher potency or a combination of drugs that exhibit a synergistic effect are favorable strategies [122].

## 8. Potential Adverse Effects of RSV

Mounting evidence suggests that RSV has health benefits and plays a role in cardiovascular protection [123]. However, human clinical studies have reported conflicting results regarding the conservative effects of RSV in various diseases and their sequelae [124]. The reasons for these conflicting findings are unknown. However, dissimilarity in the characteristics of the study subjects, the prescribed RSV dosage and the duration of RSV treatment are possible causes [125]. Numerous studies have shown that RSV has paradoxical dose-dependent effects. At low concentrations, RSV acts as an antioxidant and protects against lipid, protein and DNA damage. However, at high concentrations, RSV acts as a pro-oxidant and promotes cellular damage [4]. The optimal RSV dosage for maximizing the benefit of RSV to cardiovascular health without increasing toxicity requires further investigation [126].

## 9. Conclusions

The findings of this study suggest that RSV is a possible therapeutic option for patients with CKD with or without CVD. In this regard, RSV can influence the traditional and nontraditional CVD risk factors and alleviate the effects of uremic toxins in patients with CKD.

One of the nontraditional risk factors for CVD is uremic toxin-related cardiovascular side effects. IS causes electrical and structural remodeling in myocardial tissue, leading to atherosclerotic vascular disease. RSV treatment restores intestinal epithelial TJ proteins to increase epithelial integrity. RSV alters the gut microbiota to reduce indole levels in the intestinal lumen. Moreover, RSV inhibits hepatic SULT to reduce the production of uremic toxins such as IS.

In addition, RSV treatment markedly reduces the expression of pro-fibrotic proteins and increases the expression of the components of the ACE2/MasR axis to cause vasodilatation and reduce blood pressure. RSV inhibits the migration of vascular SMCs, which have important antiatherogenic and antiatherosclerotic effects [82,91]. Moreover, RSV exhibits anti-sclerotic activity in CKD by delaying disease progression. The development of RSV derivatives to reduce the occurrence of CV events in patients with CKD is warranted.

## Figures and Tables

**Figure 1 ijms-21-06294-f001:**
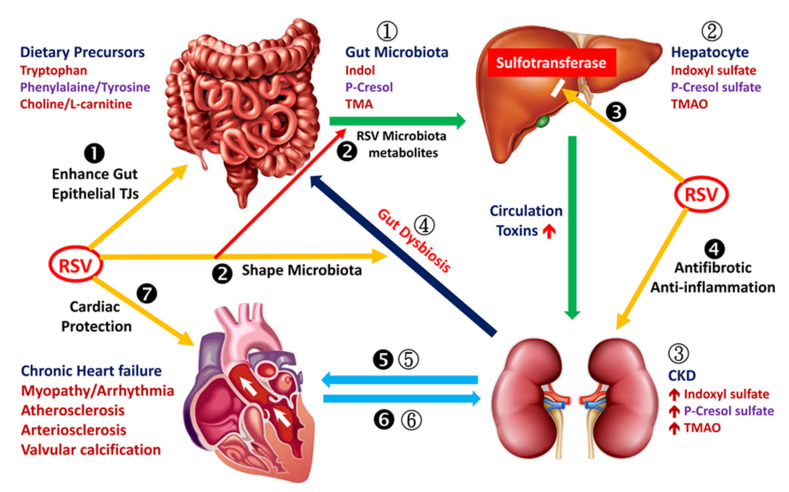
Resveratrol (RSV) suppresses uremic toxins and improves their effect on the cardiovascular system. Foods containing tryptophan, phenylalanine/tyrosine and choline/L-carnitine are metabolized into indole, *p*-cresol and trimethylamine (TMA), respectively, by gut microbiota (1). Indole, *p*-cresol and TMA are absorbed in the gut and metabolized into indoxyl sulfate (IS), *p*-cresol or *p*-cresyl sulfate (PCS) and trimethylamine-N-oxide (TMAO), respectively, in the liver (2). Thereafter, IS, PCS and TMAO enter the circulation and are excreted through the renal proximal tubules. These toxins accumulate in the body when kidney function declines (3). Chronic kidney disease (CKD) and related uremic toxins may induce changes in normal gut microbiota (4). Circulating IS and PCS and possibly TMAO cause electrical and structural remodeling of the myocardial tissue, which may lead to heart failure, atherosclerosis or arrhythmia (5). Cardiovascular abnormality accelerates the progression of renal function decline (6). Resveratrol (RSV) treatment restores intestinal epithelial tight junction proteins, thereby enhancing epithelial integrity (❶). Both RSV-altered gut microbiota and microbial metabolites of RSV contribute to decreased indole, *p*-cresol and TMA levels in the intestinal lumen (❷). RSV inhibits hepatic sulfotransferase to reduce IS and PCS production (❸); protects the kidney through its anti-inflammatory and antifibrotic effects (❹); protects the heart from chronic injury caused by IS, PCS and TMAO (❺); and prevents exacerbation of renal deterioration caused by failed cardiovascular function (❻). Resveratrol provides direct cardiac protective effects (❼).

**Figure 2 ijms-21-06294-f002:**
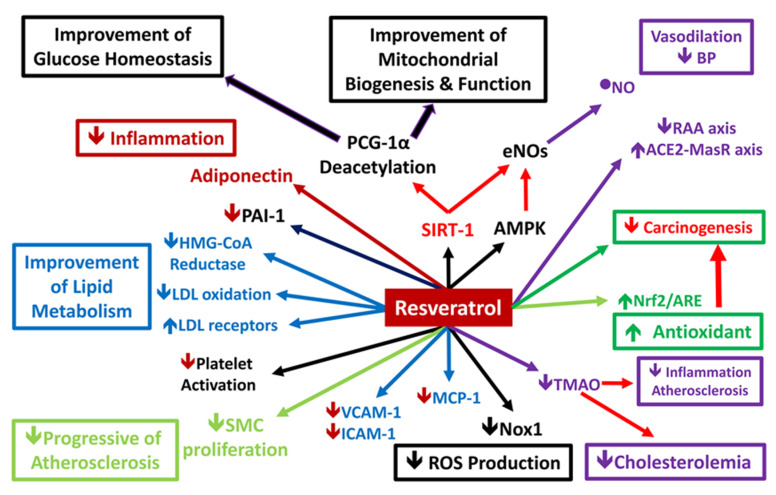
Potential mechanism of resveratrol action on metabolic syndrome. Resveratrol (RSV) improves the plasma lipid profile by reducing LDL cholesterol and triglyceride levels and increasing HDL cholesterol levels. In hepatocytes, RSV can increase HMG-CoA reductase activity and potentiate LDL receptor expression to reduce plasma LDL cholesterol. RSV also inhibits the migration of vascular smooth muscle cells, which have important antiatherogenic and antiatherosclerotic effects. RSV can also activate Nrf2, endothelial nitric oxide synthase and other antioxidant response components and attenuate the production of TNFα. RSV induces structural alterations in Keap1 protein, thereby inhibiting it from sequestering Nrf2 in the cytoplasm. Increased cytoplasmic Nrf2 translocate to the nucleus, where it binds the response elements and initiates transcription of multiple antioxidant genes^90^. Thus, RSV has antioxidative and anti-inflammatory effects. In the vascular endothelium, RSV attenuates the expression of adhesion molecules by inhibiting the NF-κB activation pathway. In vascular macrophages, RSV reduces the formation of foam cells by inhibiting nitric oxide synthase 1 and reducing monocyte chemoattractant protein-1 production. RSV prevents vascular aging by reducing the activity of the renin/angiotensin II system and stimulating the angiotensin-converting enzyme 2/mas receptor axis. RSV also provides additional anti-carcinogenic effects parallel with cardioprotective effects.

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
