# Peer review of "Influence of Resveratrol on the Cardiovascular Health Effects of Chronic Kidney Disease"

_ijms, 2020, doi:10.3390/ijms21176294_

Round 1

Reviewer 1 Report

Title: “Influence of Resveratrol on the Cardiovascular Health Effects of Chronic Kidney Disease

Authors: Jenn-Yeu Song, Ta-Chung Shen, Yi-Chou Hou, Jia-Feng Chang, Chien-Lin Lu, Wen-Chih Liu, Bo-Hau Chen, Cai-Mei Zheng and Kuo-Cheng Lu

Summary:

Natural polyphenols, such as resveratrol, have proven to be promising candidates for the treatment of chronic diseases. This review summarizes the positive effects of resveratrol on cardiovascular health in chronic kidney disease.

Comments:

A well-researched and written review article highlighting the great potential and limitations of resveratrol in relation to the cardiovascular health consequences of chronic kidney disease.

Major Points:

  1. Since this is a review and since resveratrol as a multi-targeting component also has a strong anti-carcinogenic effect via cardiovascular effects, a paragraph on the anti-carcinogenic effects of resveratrol should also be considered.
  2. Please adjust and improve the quality of the image in Figure 2.

Minor points:

  1. Once the authors have abbreviated a word or technical term, that's enough, and for the rest of MS simply use the abbreviation. Please go through MS and correct it everywhere.
  2. Page 3, line 77: “Oxidative stress” comes for the first time in line 85, please take the abbreviation.
  3. Page 3, line 104: Please write proinflammatory, like pro-inflammatory, as you write anti-inflammatory.
  4. Page 3, line 105: Tumor necrosis factor α, please take the abbreviation.
  5. Page 5, line 171: Please add additional sentence and references: Epithelisation is an important property of resveratrol, even in tumor cells. It has been shown that resveratrol also increases the expression of TJ, desmosomes in CRC (colorectal cells). Please add additional reference: Buhrmann et al, 2015, Biochemical Pharmacology 98: 51-68.
  6. Page 6, line 209: Please add additional sentence and references: Resveratrol is also activated as an intracellular signalling molecule during the anti-tumor action of resveratrol. Please add additional reference: Buhrmann et al., 2016, nutrients 2016, 8, 145.
  7. Please change everywhere “nontraditional” to “non-traditional”

Author Response

For Reviewer #1

Comments:

A well-researched and written review article highlighting the great potential and limitations of resveratrol in relation to the cardiovascular health consequences of chronic kidney disease.

Response: We are deeply honored by the time and effort you spent in reviewing this manuscript. We all learn so much from your criticism. We have revised the manuscript thoroughly according to your suggestions. The responses to your comments are below.

Major Points:  

  1. Since this is a review and since resveratrol as a multi-targeting component also has a strong anti-carcinogenic effect via cardiovascular effects, a paragraph on the anti-carcinogenic effects of resveratrol should also be considered.

Response: Thank you very much for the reviewer’s kind reminder. We had made the revision according to the reviewer’s suggestion. A new paragraph regarding the anti-carcinogenic effect via cardiovascular effects was added as paragraph 6 in the revised manuscript.

(Line 347-361 of revised manuscript)

  1. Please adjust and improve the quality of the image in Figure 2.

Response: Thank you very much for reviewer’s valuable advice. We had made the necessary rectification according to the reviewer’s comments. The revised Figure 2 will be easier for readers to read. We had added more information in the revised Figure 2.

(Line 296-312 of revised manuscript).

Minor points:

  1. Once the authors have abbreviated a word or technical term, that's enough, and for the rest of MS simply use the abbreviation. Please go through MS and correct it everywhere.

Response: Thank you very much for the reviewer’s kind reminder. We had made the revision according to the reviewer’s suggestion. We have gone through MS and correct it everywhere.

  1. Page 3, line 77: “Oxidative stress” comes for the first time in line 85, please take the abbreviation.

Response: We had taken the abbreviation “OS” in place of “oxidative stress” after line 85.

  1. Page 3, line 104: Please write pro-inflammatory, like pro-inflammatory, as you write anti-inflammatory.

Response: We write pro-inflammatory in place of pro-inflammatory as reviewer’s suggestions.

  1. Page 3, line 105: Tumor necrosis factor α, please take the abbreviation.

Response: We write TNF-α in place of “tumor necrosis factor α” as reviewer’s suggestions.

  1. Page 5, line 171: Please add additional sentence and references: Epithelisation is an important property of resveratrol, even in tumor cells. It has been shown that resveratrol also increases the expression of TJ, desmosomes in CRC (colorectal cells). Please add additional reference: Buhrmann et al, 2015, Biochemical Pharmacology 98: 51-68.

Response: Thank you very much for the reviewer’s kind reminder. We had made the revision according to the reviewer’s suggestion. We added more sentences and references as reviewer’s suggestions. (Revised manuscript reference 39)

(Line 170-172, Revised manuscript: Paragraph: 3.1.1)

  1. Page 6, line 209: Please add additional sentence and references: Resveratrol is also activated as an intracellular signalling molecule during the anti-tumor action of resveratrol. Please add additional reference: Buhrmann et al., 2016, nutrients 2016, 8, 145.

Response: Thank you very much for reviewer’s valuable advice. We had made the necessary rectification according to the reviewer’s comments. We added a paragraph to describe the significant role of RSV on anti-cancer activity through targeting Sirt1 and suppression of NF-κB activation.

(Line 359-360, Revised manuscript: Paragraph 6)

  1. Please change everywhere “nontraditional” to “non-traditional”

Response: Thank you very much for the reviewer’s valuable comments. We had made the necessary revision according to the reviewer’s opinion.

Reviewer 2 Report

Comments:

This manuscript describes a study about “influence of RSV on the CVD effects of CKD. The authors have published some relevant research, and the research in this field is quite complete. This paper expound the beneficial effects of RSV on biological, pathophysiological, and clinical responses in patients with CVD and CKD. This manuscript is well written. The discussion is clearly described and thorough. Therefore, I do recommend this manuscript for publication in IJMS.

Author Response

For Reviewer #2

Comments:

This manuscript describes a study about “influence of RSV on the CVD effects of CKD. The authors have published some relevant research, and the research in this field is quite complete. This paper expound the beneficial effects of RSV on biological, pathophysiological, and clinical responses in patients with CVD and CKD. This manuscript is well written. The discussion is clearly described and thorough. Therefore, I do recommend this manuscript for publication in IJMS.

Response: We are deeply honored by the time and effort you spent in reviewing this manuscript. We all learn so much from your criticism. We have revised the manuscript thoroughly according to journal guideline.